# In silico and in vitro studies on the anti-cancer activity of andrographolide targeting survivin in human breast cancer stem cells

Septelia Inawati Wanandi[1,2,3]*, Agus Limanto[4], Elvira Yunita[4], Resda Akhra Syahrani[3,4], Melva Louisa[5], Agung Eru Wibowo[6], Sekar Arumsari[3]

1 Department of Biochemistry and Molecular Biology, Faculty of Medicine, Universitas Indonesia, Jakarta, Indonesia, 2 Center for Hypoxia and Oxidative Stress Studies, Faculty of Medicine, Universitas Indonesia, Jakarta, Indonesia, 3 Molecular Biology and Proteomic Core Facilities, Indonesian Medical Education and Research Institute, Faculty of Medicine, Universitas Indonesia, Jakarta, Indonesia, 4 Master Program in Biomedical Sciences, Faculty of Medicine, Universitas Indonesia, Jakarta, Indonesia, 5 Department of Pharmacology, Faculty of Medicine, Universitas Indonesia, Jakarta, Indonesia, 6 Laboratory for Development of Industrial Agro and Biomedical Technology (LAPTIAB), Agency for the Assessment and Application of Technology (BPPT), Serpong, Tangerang Selatan, Indonesia

* septelia.inawati@ui.ac.id, septelia@gmail.com

**Data Availability Statement:** All relevant data are within the paper and its Supporting information files.

## Abstract

Breast cancer stem cells (BCSCs) express high levels of the anti-apoptotic protein, survivin. This study aimed to discover a natural active compound with anti-cancer properties that targeted survivin in human breast cancer stem cells. From the seven examined compounds, andrographolide was selected as a lead compound through in silico molecular docking with survivin, caspase-9, and caspase-3. We found that the affinity between andrographolide and survivin is higher than that with caspase-9 and caspase-3. Human CD24-/CD44+ BCSCs were treated with andrographolide in vitro for 24 hours. The cytotoxic effect of andrographolide on BCSCs was compared to that on human mesenchymal stem cells (MSCs). The expression of survivin, caspase-9, and caspase-3 mRNA was analyzed using qRT-PCR, while Thr34-phosphorylated survivin and total survivin levels were determined using ELISA and Immunoblotting assay. Annexin-V/PI flow cytometry assays were performed to evaluate the apoptotic activity of andrographolide. Our results demonstrate that the $CC_{50}$ of andrographolide in BCSCs was 0.32mM, whereas there was no cytotoxic effect in MSCs. Moreover, andrographolide decreased survivin and Thr34-phosphorylated survivin, thus inhibiting survivin activation and increasing survivin mRNA in BCSCs. The apoptotic activity of andrographolide was revealed by the increase of caspase-3 mRNA and protein, as well as the increase in both the early and late phases of apoptosis. In conclusion, andrographolide can be considered an anti-cancer compound that targets BCSCs due to its molecular interactions with survivin, caspase-9, and caspase-3, which induce apoptosis. We suggest that the binding of andrographolide to survivin is a critical aspect of the effect of andrographolide.

**Funding:** "This work was funded by the PDUPT Grant (NKB-1564/UN2.R3.1/HKP.05.00/2019) from the Ministry of Research, Technology and Higher Education of the Republic of Indonesia. The funders had no role in study design, data collection and analysis, decision to publish, or preparation of the manuscript."

**Competing interests:** The authors have declared that no competing interests exist.

## Introduction

Breast cancer, like other solid cancers, contains cell populations known as breast cancer stem cells (BCSCs) which have tumorigenic, pluripotent, and self-renewal properties. The existence of BCSCs may be responsible for the occurrence of metastasis, therapy resistance, and the recurrence of breast cancer. Human breast cancer stem cells can be recognized by several surface antigen markers, such as CD24-/CD44+ [1].

Survivin is an inhibitor of the apoptosis protein (IAP) family that blocks the intrinsic apoptosis pathway through binding with either caspase-9 or caspase-3. The phosphorylation of survivin on Thr34 residue increases the binding of this protein to initiator caspase-9, leading to the inhibition of caspase-9 and caspase-3 interaction [2–4]. This protein is rarely present in normal cells but is highly expressed in cancer cells [5, 6]. Our previous study demonstrated that human BCSCs (CD24-/CD44+) have higher viability compared to their counterpart non-BCSCs (CD24-/CD44-) due to their high expression of survivin [7]. It has previously been reported that survivin expression is regulated by cancer stem cell signaling pathways such as Wnt [8]. Therefore, targeting survivin inhibition is a promising strategy for apoptosis-based cancer therapy, particularly in cancer stem cells. This could possibly be done by reducing the expression of survivin in human breast cancer stem cells (BCSCs) using natural active compounds from medicinal plants.

The aim of this study was to discover a natural active compound that targeted survivin for anti-cancer treatment. We first performed an *in silico* analysis, with particular focus on the molecular docking simulation, to determine one lead compound among seven natural active compounds from medicinal plants—curcumin, rocaglamide, α-mangostin, 6-gingerol, 8-gingerol, 10-gingerol, and andrographolide, the latter of which has the strongest binding activity towards the survivin target protein. These selected active compounds are found among medicinal plants such as *Curcuma domestica*, *Curcuma xantorrhiza*, *Aglaia sp.*, *Andrographis paniculata* and *Garcinia mangostana* that are endemic in Southeast Asian countries, including Indonesia [9–11]. Therefore, utilization of these compounds for anticancer offers a potential benefit, both economically and clinically. All of these active compounds are widely used in cancer research due to their potential anti-cancer activity [12–20]. Furthermore, the molecular ligand-protein and protein-protein interactions between the selected lead compound, survivin, caspase-9, and caspase-3 were analyzed *in silico*. Following this analysis, an *in vitro* study was conducted using andrographolide to evaluate its effect on the expression of survivin, caspase-9, and caspase-3, as well as its impact on the apoptosis of human BCSCs. Further studies are required to confirm the *in vivo* therapeutic effect of andrographolide as a novel anti-cancer treatment targeting breast cancer stem cells.

## Materials and methods

### *In silico* study

**Selection of lead compound.** As presented in Fig 1, a lead compound was selected via molecular docking simulation from among seven natural active compounds which have anti-cancer properties. Visualization of the 3D structures of the seven natural active compounds was first performed using ChemDraw Ultra 13. The active compounds were optimized by adding hydrogen atoms and bonds between atoms, followed by the addition of potential setups using MMFF94x parameters; the calculation of the drug-like properties of these compounds was performed using MOE 2010.10 [21–23]. Following this process, Lipinski's Rule of Five was applied to verify the druglikeness property of these compounds [24–26].

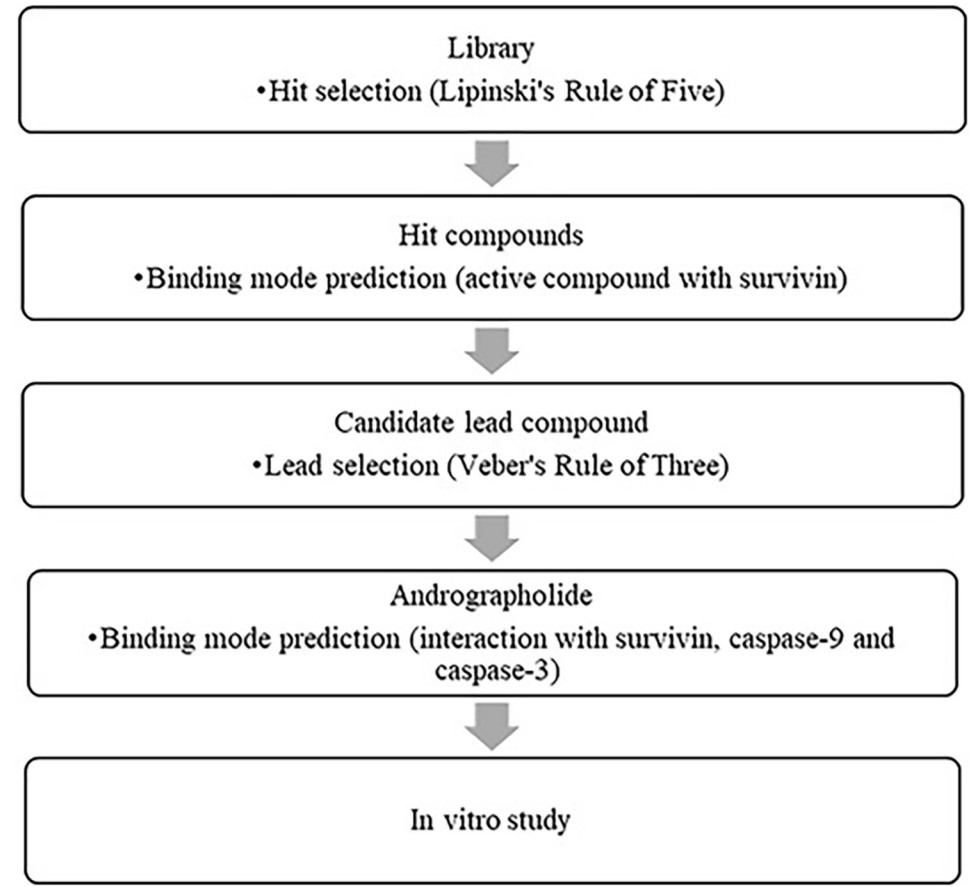

**Fig 1. Flowchart of research methodology.**

The seven active compounds were further analyzed for their binding potential with survivin. The amino acid sequence of the human survivin protein was retrieved from the Protein Data Bank (http://www.ncbi.nlm.nih.gov/protein/). Then an amino acid sequence alignment was performed through SWISS-MODEL to obtain a proper and accurate 3D model of survivin from the RCSB Protein Data Bank [27]. Afterward, the 3D structure of survivin was evaluated and validated using a Ramachandran plot [28]. The positions of the hydrogen atoms, the side chain atoms, and the geometries of the protein model were optimized using CHARMM [21]. The binding site for an active compound with each protein target was defined using the MOE-Alpha Site Finder module. The conformation generation for an active compound was performed using fixed bond lengths and bond angles. The parameter used for poses generation was the London dG score [19]. The top 30 poses were chosen from 3000 poses for each ligand. These poses were further optimized using MMFF94x with a GB/VI model. The two compounds with the highest values were selected and further analyzed using Veber's rules to obtain a lead compound [25, 26]. Selection of the lead compound was performed by analyzing the residue interactions specifically targeted to the Thr34 site of survivin and the hydrogen bonds and electrostatic interactions generated from the molecular docking of both hit compounds with survivin.

**Protein-protein binding prediction.** To analyze the protein-protein binding prediction, we determined the molecular modeling of caspase-9 and caspase-3 as aforementioned for

survivin. The protein–protein binding mode prediction was performed using PatchDock to predict the survivin–caspase-9, survivin–lead compound-caspase-9, caspase-9–caspase-3, and caspase-9–lead compound–caspase-3 interactions [29, 30]. Each protein was considered as a piece of a puzzle and the interaction between two proteins was analyzed by matching these puzzle pieces. Then, a calculation of the binding energy between two proteins and the ligand was performed. The results of the protein-protein binding mode prediction were refined using FireDock (Fast Interaction in Molecular Docking Refinement) [31, 32]. The results of the Fire-Dock calculation were determined as global binding energy including the hydrogen bonds between atoms (Atomic Contact Energy) and van der Waals interactions.

## *In vitro* study

This study received the ethical approval from the Health Research Ethics Committee (No. 599/ PT02.FK/ETIK/2011), Faculty of Medicine, Universitas Indonesia—Cipto Mangunkusumo Hospital, according to the Declaration of Helsinki. Breast cancer tissue has been collected from a triple negative breast cancer patient underwent surgery. Written informed consent form has been signed by the patient prior to collecting the specimen. Mesenchymal stem cells (MSCs) were obtained from Prof. Endang Winiati Bachtiar (Laboratory of Oral Biology Faculty of Dentistry, Universitas Indonesia) who has isolated the MSCs from dental pulp of a patient underwent tooth extraction with a written informed consent signed by the patient and approved by the Ethics Commission of Research and Community Service Institution at Bogor Agricultural Institute (No. 05–2015 IPB). This study was conducted according to the ethical principles.

**Cell culture.** Human BCSCs (CD24-/CD44+) have been isolated from the primary culture of breast cancer cells (Granted patent from the General Directorate of Intellectual Property Right, Ministry of Law and Human Right, Republic of Indonesia No. IDP00056854) and characterized for their pluripotency and tumorigenicity (Granted patent from the General Directorate of Intellectual Property Right, Ministry of Law and Human Right, Republic of Indonesia No. IDP000060309). The human BCSCs and MSCs have been used in the previous studies [33, 34]. Both cells were grown in high glucose Dulbecco's modified Eagle's medium (DMEM)/F12 (Gibco) without fetal bovine serum (FBS) and supplemented with 1% Penicillin-Streptomycin under standard conditions (5% CO2 at 37°C). Whenever the cultures reached 80–90% confluence, the cells were harvested with 0.25% Trypsin and 1% EDTA.

**Andrographolide treatment.** About $1x10^5$ cells per well were grown in a 24-well plate with 800µl DMEM/F-12 medium under standard conditions for 24 hours. Andrographolide powder (Sigma Aldrich; Product No. 365645; Lot No. MKBN8939V) was diluted in 0.01% DMSO. Afterwards, cells were treated with 200µl DMSO-diluted andrographolide at final concentrations of 0.03mM, 0.075mM, 0.15mM, 0.30mM and 0.60mM, respectively. The control was cells treated with 0.01% DMSO (vehicle). After harvesting, viable cells were counted in order to determine the cytotoxic concentration-50 ($CC_{50}$) using a trypan blue dye exclusion assay with automated cell counter (LunaTM), total RNA and protein were isolated for expression analysis, and cells were used for an apoptosis assay.

**Determination of survivin, caspase-9 and caspase-3 mRNA expression levels in human BCSCs.** To analyze the effect of andrographolide on survivin, caspase-9, and caspase-3 mRNA expression in human BCSCs, cells were treated with various concentrations (0.075mM, 0.15mM, 0.3mM and 0.6mM) of andrographolide under standard culture conditions for 24 hours. The RNA was extracted from harvested cells using a Tripure RNA isolation kit (Roche, Switzerland). The total RNA concentration was determined using spectrophotometry at λ 260 nm. About 100ng of total RNA was used as a template for quantitative RT-PCR

**Table 1. Specific primers of 18S rRNA, caspase-9, caspase-3 and survivin genes.**

| Gene | Primers | Sequences | Product Length |
|---|---|---|---|
| **18S rRNA** | *Forward* | 5'- AAACGGCTACCACATCCAAG -3' | 153 |
| | *Reverse* | 5'- CCTCCAATGGATCCTCGTTA -3' | |
| **Caspase-9** | *Forward* | 5'- TTGGTGATGTCGAGCAGAAA -3' | 164 |
| | *Reverse* | 5'- GGCAAACTAGATATGGCGTC -3' | |
| **Caspase-3** | *Forward* | 5'-TGAGGCGGTTGTAGAAGAGTT-3' | 155 |
| | *Reverse* | 5'-CACACCTACCGATAACCAGAG-3' | |
| **Survivin** | *Forward* | 5'- GCCAGATGACGACCCCATAGAGGA-3' | 273 |
| | *Reverse* | 5'- TCGATGGGCACGGCGCACTTT-3 | |

using a KAPA SYBR® FAST Universal One-Step qRT-PCR Kit (Kapa Biosystems, Merck, Germany), as described in the manufacturer's protocol. The reactions used were cDNA synthesis for 10 minutes at 50°C, inactivation of reverse transcriptase for 5 minutes at 95°C, and PCR cycles (40 cycles) for 10 seconds at 95°C, 30 seconds at 59°C (for 18Sr RNA) or at 61°C (for survivin), and 30 seconds at 72°C. 18S rRNA was used as a reference gene.

The primers for survivin, caspase-9, caspase-3, and 18S rRNA cDNA amplification were designed using the Primer-Blast program and NCBI Gene Bank [NM_001168.2, NM_01278054.1, NM_004346.4, and NC_000021.9, respectively]. Sequences of primers used in this study are shown in Table 1. Aquabidest was used as a non-template control (NTC) to reduce the chances of a false positive result. The level of mRNA expression in andrographolide-treated cells was determined using Livak's formula and normalized to DMSO-treated cells as a control [35].

**Determination of total and phosphorylated survivin level in human BCSCs.** The total protein from human BCSCs was isolated using a RIPA lysis buffer (Santa Cruz Biotechnology, US & Canada). Harvested cells were incubated for 30 minutes and vortexed for 10–15 seconds every 10 minutes. Afterward, cells were centrifuged at 16000 g for 20 minutes. The total protein concentration was measured using a Nanodrop Thermo Scientific Varioskan Flash® (Thermo Scientific) spectrophotometer at a wavelength of 280 nm.

The total (both phosphorylated and non-phosphorylated) survivin and Thr34 phosphorylated survivin levels were analyzed using PathScan® Total Survivin and PathScan® Phospho-Survivin (Thr34) Sandwich ELISA Kits (Cell Signaling Technology, USA), according to the manufacturer's protocol. To determine the total survivin and phosphorylated protein levels, we measured the absorbances using a Varioskan Flash Multimode Reader (Thermo Fisher Scientific, USA) at a wavelength of 450 nm and divided by the total protein concentration.

To verify the ELISA result of survivin and Thr34-phosphorylated suvivin protein levels, we also performed Immunoblotting assay. Protein lysates were separated using 12% SDS polyacrylamide gel electrophoresis in equal amounts and transferred onto nitrocellulose membrane using wet blot technique, and the membrane were soaked in blocking buffer using 5% BSA and skimmed milk for 1 h in room temperature. Membrane were horizontally cut into two parts at 30 kDa level and then incubated with anti-β-actin antibody (Cell Signaling Technology, USA), anti-survivin antibody (Cell Signaling Technology, USA), anti-survivin (phospho T34) antibody (Abcam, USA), anti-caspase-9 antibody (Cell Signaling Technology, USA), or anti-caspase-3 (Abcam, USA) overnight at 4°C. Protein were incubated in HRP-conjugated secondary antibody (Santa Cruz Biotechnology, Inc., Texas, USA) and visualized by ECL as the HRP substrate (Optiblot ECL Substrate Kit, Abcam, USA).

**Determination of apoptosis activity in human BCSCs.** An apoptosis assay was conducted using an Annexin V-FITC Apoptosis Detection Kit (Abcam, UK) and Propidium

Iodide staining (Abcam, HK), according to the manufacturer's protocol. A total of $5 \times 10^5$ cells were treated with andrographolide at concentrations of 0.075 mM, 0.15 mM, 0.3 mM, and 0.6 mM. After a 24-hour incubation period, cells were harvested and resuspended in a 500 µl binding buffer solution. Following this step, 5 µl Annexin V-FITC and 5 µl propidium iodide were added to the cell suspension. The mixture of cell suspension was then stored in the dark for 5 minutes and quantified using BD FACS Verse flow cytometry (Becton, Dickinson).

**Statistical analysis.** Data were analyzed using SPSS software version 26. Data is presented as mean ± standard deviation (SD) of three independent experiments with three replicates. Coefficient of variation (COV) is calculated as a percentage ratio of the SD to the mean of expression level. The COV value expresses the homogeneity of data distribution. Normality tests performed on all experiment data were Kolmogorof-Smirnov test and the Levene homogeneity test. Since the data distribution is normal and homogenous, the analysis was continued with the parametric test of one-way analysis of unpaired variance (one-way ANOVA, unrelated) followed by the Tuckey test to show the differences between groups.

## Results

### Andrographolide is the lead compound targeted to survivin

In this study, we performed a physicochemical analysis of seven natural active compounds as shown in Table 2. According to the criteria of Lipinski's Rule of Five, all active compounds have the property of druglikeness [23, 24].

The molecular model of survivin as a target protein was obtained from the NCBI database (Accession number AAC51660). After alignment with SWISS-MODEL, the 3D structure of survivin was obtained from the RCSB Protein Data Bank (PDB ID 1E31) with GMQE score 0.99 and RMSD value 2.9Å. The Ramachandran plot of 1E31 showed that 250 residues (92.6%) lie in the favored region, 17 residues (6.3%) lie in the allowed region, and 3 residues (1.1%) lie in the outlier region.

Binding mode predictions between all active compounds and survivin were analyzed; the results are shown in Table 3. We found that andrographolide and rocaglamide were the two compounds with the highest affinity for survivin.

Furthermore, the interactions of andrographolide and rocaglamide targeting the phosphorylation site of survivin, Thr34, were analyzed. As demonstrated in Table 4 and Fig 2, andrographolide specifically interacts with the Thr34 phosphorylation site of survivin, whereas rocaglamide interacts with the Ile19 site. Therefore, we selected andrographolide as the lead compound targeted to survivin.

### Molecular interaction of andrographolide with survivin, caspase-9, and caspase-3

The result obtained from the blind docking of andrographolide and survivin indicated that andrographolide also interacts with seven other survivin residues with a slightly higher affinity than with the Thr34 phosphorylation site, as shown in Table 5.

**Table 2. Physicochemical data of seventh active compounds.**

|  | Andrographolide | Curcumin | 6-Gingerol | 8-Gingerol | 10-Gingerol | Mangostin | Rocaglamide |
|---|---|---|---|---|---|---|---|
| **Mr** | 350.45 | 368.38 | 294.39 | 322.44 | 350.49 | 410.46 | 505.56 |
| **logP** | 1.963 | 3.370 | 3.234 | 4.014 | 4.794 | 5.166 | 3.674 |
| **donor H** | 3 | 2 | 2 | 2 | 2 | 3 | 2 |
| **acceptor H** | 4 | 6 | 4 | 4 | 4 | 5 | 7 |
| **TPSA** | 86.99 | 93.06 | 66.76 | 66.76 | 66.76 | 96.22 | 97.69 |

**Table 3. Energy values obtained in molecular docking calculations with survivin.**

| Compounds | ΔG binding (kcal/mol) | pKi |
|---|---|---|
| Rocaglamide | -14.2737 | 10.063 |
| Andrographolide | -10.5464 | 7.435 |
| 8-gingerol | -9.9958 | 7.047 |
| α-mangostin | -9.2930 | 6.551 |
| 10-gingerol | -9.2800 | 6.542 |
| Curcumin | -9.0839 | 6.404 |
| 6-gingerol | -8.8067 | 6.208 |

**Table 4. Interaction of andrographolide and rocaglamide with survivin on Thr34 residue.**

| Compounds | ΔG binding (kcal/mol) | Residue Interaction | Hydrogen Bond | Electrostatic Interaction |
|---|---|---|---|---|
| Rocaglamide | -5.3851 | Ile 19 | 1 | 7 |
| Andrographolide | -8.2617 | Thr34; Glu36 | 2 | 4 |

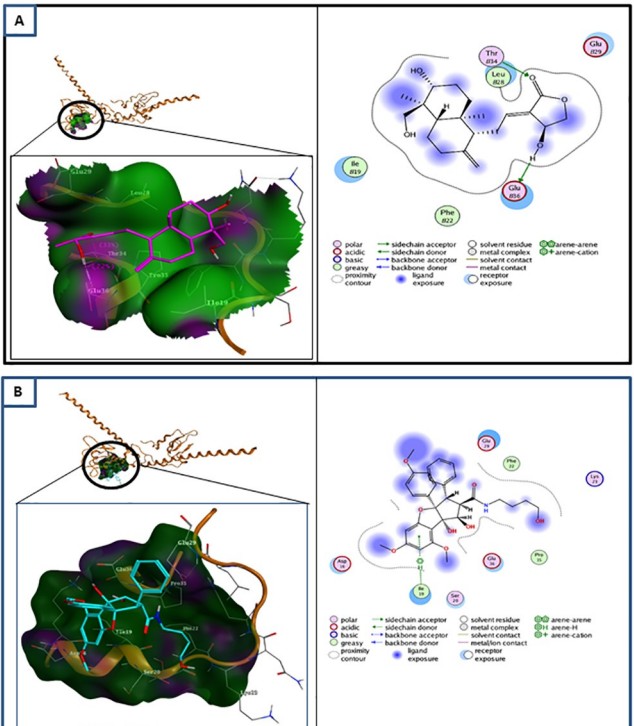

**Fig 2. Molecular docking of andrographolide and rocaglamide with survivin. (A)** (left) Molecular docking of andrographolide (purple) at the Thr34 phosphorylation site of survivin (brown). At phosphorylation site Thr34, andrographolide (H-donor) interacts with Thr34 and Glu36 residues. (right) Ligand interaction between andrographolide and survivin. **(B)** (left) Molecular docking of rocaglamide (blue) with survivin (brown) at phosphorylation site Thr34. It interacts (pi-H/hydrogen acceptor) with Ile19 residue. (right) Ligand interaction between rocaglamide and survivin. Rocaglamide has no interaction with Thr34 but reacts with Ile19 residue.

**Table 5. Molecular docking between andrographolide and survivin.**

| | ΔG binding (kcal/mol) | Residue Interaction | Hydrogen Bond | Electrostatic Interaction |
|---|---|---|---|---|
| **Blind-docking** | -10.5464 | 5; 102; 105; 106; 109; 110; 113 | 1 | 6 |
| **Thr34** | -8.2617 | 34; 36 | 2 | 4 |

Following the molecular interaction study with survivin, we also analyzed the molecular docking of andrographolide with other proteins in the intrinsic apoptosis pathway using molecular models of cleaved caspase-9 (PDB ID 2AR9) and procaspase-3 (PDB ID 1QX3), as well as the energy binding of protein-ligand and protein-protein interactions between survivin, active caspase-9, and procaspase-3 using protein-protein binding mode predictions, as shown in Table 6. Fig 3 demonstrates that andrographolide interacts with active caspase-9 through hydrogen bonds at Asp228, Lys 278, and Lys409, as well as with procaspase-3 through Phe275 amino acid residues.

Table 6 demonstrates the binding energy of andrographolide with survivin and caspase-9, survivin and caspase-3, and caspase-9 and caspase-3. The interaction of andrographolide with survivin (blind docking) had the highest affinity (ΔG binding = -10.5464 kcal/mol) compared to its interactions with active caspase-9 (-8.2861 kcal/mol) and procaspase-3 (-8.0742 kcal/mol). Interestingly, the interaction of survivin with caspase-9 was inhibited in the presence of andrographolide, which can be predicted from the increase of binding energy from -18.3 to -7.13 kcal/mol. Similar to the aforementioned effect, andrographolide also inhibited the interaction of survivin with caspase-3. On the other hand, the binding energy for caspase-9 and caspase-3 interaction was reduced from -3.59 to -20.52 kcal/mol in the presence of andrographolide, implying that andrographolide could enhance the affinity of caspase-9 and caspase-3.

## Andrographolide has cytotoxic activity in human BCSCs, but not in MSCs

To determine the cytotoxicity of andrographolide, the cytotoxicity concentration-50 ($CC_{50}$) of andrographolide in human BCSCs was compared with that in human MSCs. The results showed that the viability of BCSCs decreased linearly with increasing andrographolide concentrations. The $CC_{50}$ value of andrographolide in BCSCs is 0.32mM (Fig 4A). Moreover, our results showed that applying andrographolide to BCSCs could reduce the viability of BCSCs in a dose-dependent manner. Compared to the control, BCSC viability was significantly decreased after andrographolide treatment at concentrations of 0.3 mM (p<0.01) and 0.6 mM

**Table 6. Binding energy of molecular interactions of andrographolide with survivin, caspase-9, and caspase-3.**

| Molecular interaction | Binding Energy (kcal/mol) |
|---|---|
| **survivin–andrographolide** | -10.5464 |
| **caspase-9 –andrographolide** | -8.2861 |
| **caspase-3 –andrographolide** | -8.0742 |
| **survivin–caspase-9** | -18.30 |
| **survivin–andrographolide–caspase-9** | -7.13 |
| **survivin–caspase-3** | -20.12 |
| **survivin–andrographolide–caspase-3** | -6.02 |
| **caspase-9 –caspase-3** | -3.59 |
| **caspase-9 –andrographolide–caspase-3** | -20.52 |

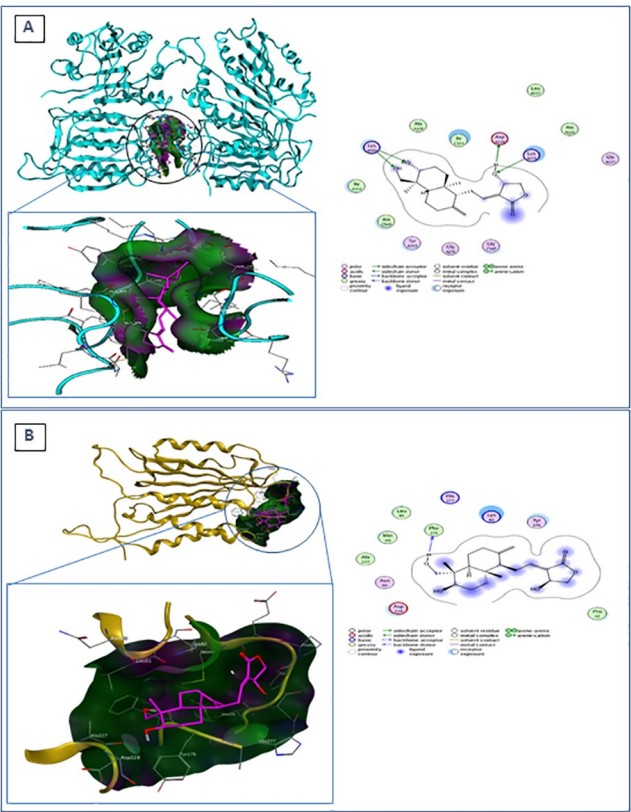

**Fig 3. Molecular docking of andrographolide with caspase-9 and caspase-3. (A)** (left) Molecular docking of andrographolide (purple) with caspase-9 (blue). (right) Ligand interaction between andrographolide and caspase-9. Andrographolide interacts with Asp228 as an H-donor and with Lys278 and Lys409 as an H-acceptor. **(B)** (left) Molecular docking of andrographolide (purple) with caspase-3 (yellow). It acts as an H-donor to Phe275. (right) Ligand interaction between andrographolide and caspase-3.

(p<0.001), respectively. On the other hand, andrographolide treatment on MSCs had no significant effect on cell viability, even at $CC_{90}$ (0.60mM) of andrographolide to BCSCs (Fig 4B).

Cell morphology was assessed after 24-hour treatment with andrographolide. Fig 4C demonstrates that the untreated BCSCs appeared healthy and formed mammospheres, but BCSCs treated with 0.6 mM andrographolide lost their ability to form mammospheres and grew as single cells (Fig 4D). In contrast to BCSCs, MSC morphology was not affected to andrographolide treatment until reaching a concentration of 0.6 mM.

## Andrographolide reduces phosphorylated and total survivin protein levels and affects the mRNA expression of survivin, caspase-9, and caspase-3

To analyze the effects of andrographolide on the mRNA expression of survivin, caspase-9, and caspase-3, we treated BCSCs with various concentrations of andrographolide for 24 hours. This study demonstrates that the addition of 0.6 mM andrographolide to BCSCs significantly increased the expression of survivin mRNA (p<0.001; Fig 5A). Interestingly, andrographolide-treated BCSCs did not demonstrate any significant change in caspase-9 expression. On the other hand, the caspase-3 mRNA expression of BCSCs was significantly increased when treated with 0.075 mM (p<0.001) and 0.15 mM (p<0.001) of andrographolide, but the expression was reduced to normal levels when treated with more than 0.15 mM of andrographolide.

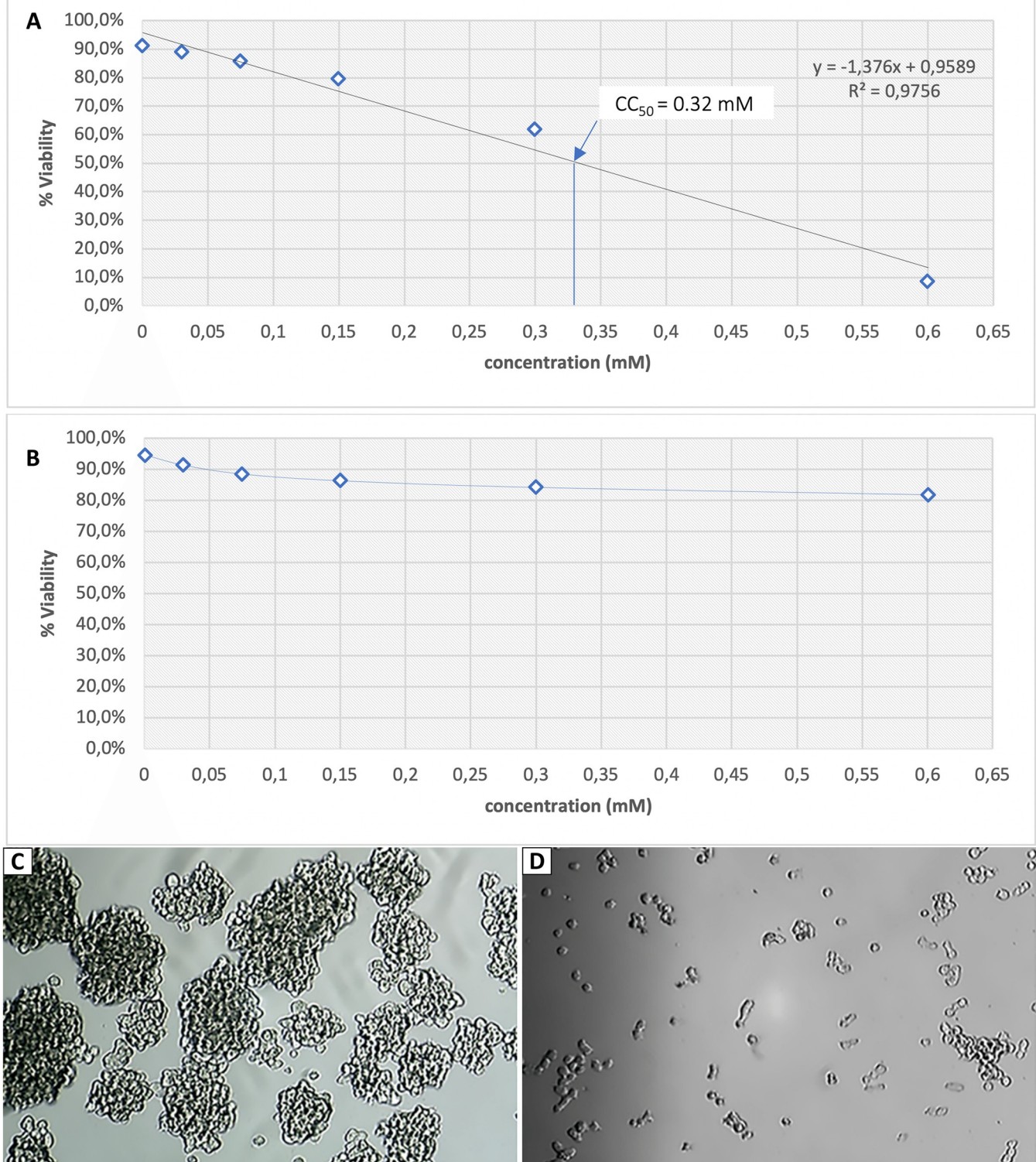

**Fig 4. Cytotoxic activity of andrographolide.** (A) $CC_{50}$ of andrographolide in BCSCs; (B) $CC_{50}$ of andrographolide in MSCs. Andrographolide was diluted with DMSO 0.01% to obtain final concentrations of 0.075, 0.15, 0.3, and 0.6 mM, respectively. C: control cells treated with DMSO 0.01%. Percentage of cell viability is shown as the mean ± SD from three independent experiments ([**] $p < 0.01$; [***] $p < 0.001$ compared to control). Following 24-h andrographolide treatment, BCSC morphology was observed under an inverted microscope (OPTIKA Srl, Ponteranica, Italy) with 100x magnification. (C) BCSC morphology untreated with andrographolide; (D) BCSC morphology treated with 0.6 mM andrographolide.

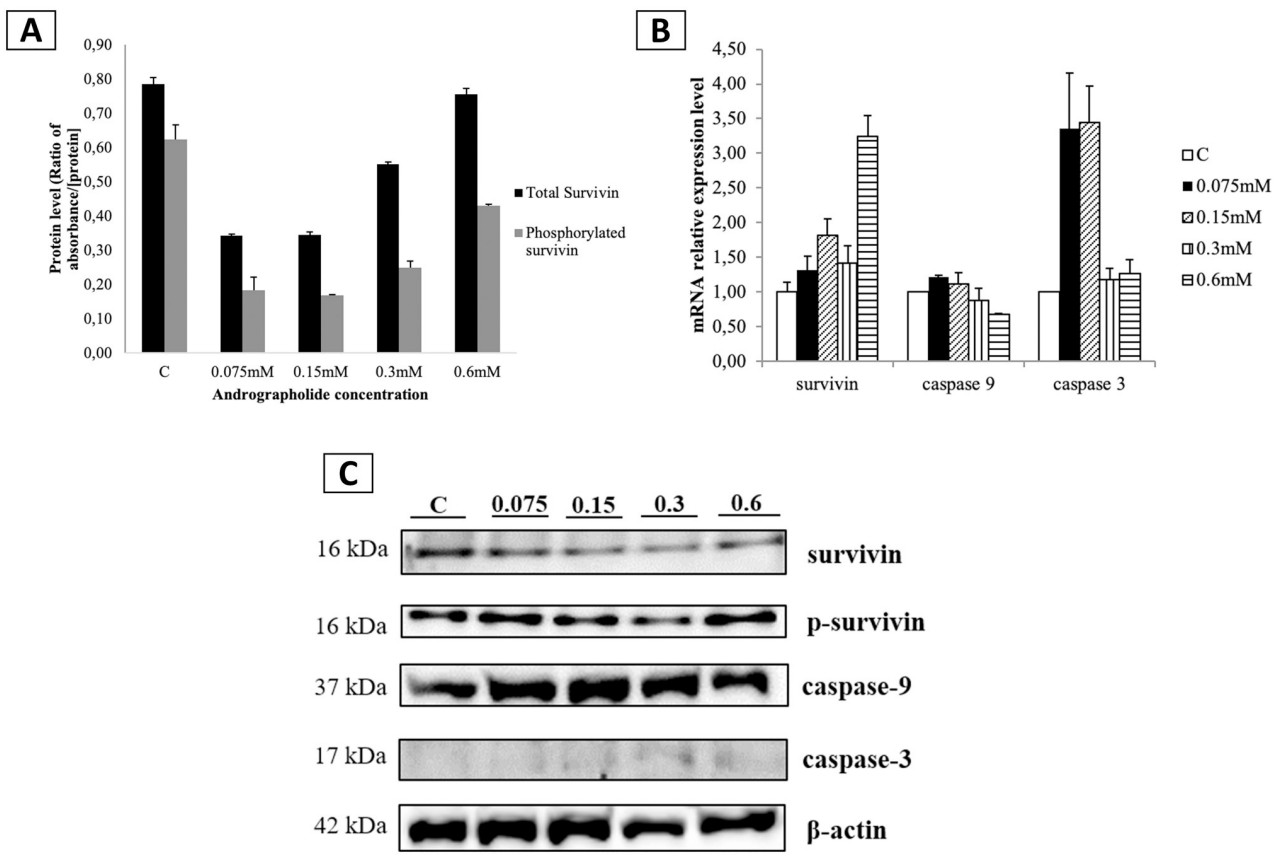

**Fig 5. Effect of andrographolide on the expression levels of survivin, caspase-9, and caspase-3 in BCSCs.** (A) mRNA expression levels of survivin, caspase-9, and caspase-3 in BCSCs treated with various concentrations of andrographolide; (B) Protein levels of total survivin and Thr34-phosphorylated survivin in BCSCs treated with various concentrations of andrographolide analyzed using ELISA; (C) Immunoassay results of survivin, Thr34-phosphorylated survivin, active caspase-9, and active caspase-3; C: control cells treated with DMSO 0.01% (vehicle). Data (A) and (B) were shown as the mean ± SD from three independent experiments. One-way ANOVA followed by Tuckey's multiple comparison tests were used to determine mean differences between groups. Statistical significance is shown in the figure as follows: **p<0.01 and ***p<0.001 compared to control.

In Fig 5B, BCSCs treated with andrographolide concentrations of 0.075 mM, 0.15 mM, and 0.3 mM showed a significant decrease (p<0.001) in total and Thr34-phosphorylated survivin protein levels compared to the control. However, the 0.6 mM andrographolide treatment did not significantly affect the levels of either total or phosphorylated survivin. We also verified the ELISA result by performing the immunoblot assay. As shown in Fig 5C, both survivin and phosphorylated survivin were decreased after andrographolide treatment until 0.3 mM, and this inhibition effect of andrographolide was reduced at concentration of 0.6 mM which may be due to the upregulated survivin mRNA expression (Fig 5A). Fig 5C also demonstrates that andrographolide increased the caspase-9 and caspase-3 protein levels.

## Andrographolide induces apoptosis of human BCSCs

To analyze the effect of andrographolide on the apoptosis of human BCSCs, we applied FIT-C-Annexin V and propidium iodide (PI) double staining for a flow cytometry assay. Fig 6A and 6B show an increasing cell death ratio between early and late apoptosis in line with the increasing concentration of andrographolide. Interestingly, less than 0.5% of the cell population underwent the necrotic phase (Q1 quadrant; Fig 6A) in all concentrations of andrographolide treatment.

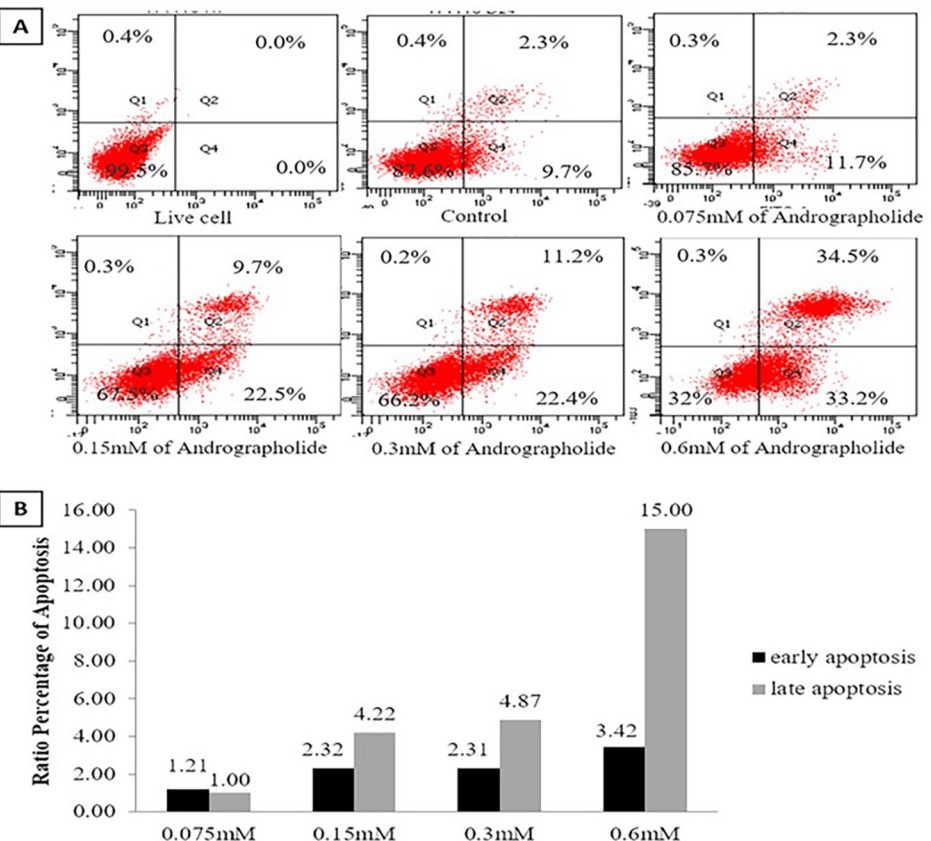

**Fig 6. Apoptosis analysis of andrographolide-treated BCSCs using flow cytometry.** (A) Cells were double stained with Annexin V-FITC and PI after treatment with andrographolide in various concentrations for 24 hours; (B) Ratio of early and late apoptosis compared to control; C: control cells treated with DMSO 0.01%.

When comparing the early and late phases of apoptosis, the increasing number of cells in the late phase is more pronounced than in the early phase (Fig 6B). The 0.075 mM andrographolide treatment did not significantly induce apoptosis in BCSCs. On the other hand, andrographolide concentrations of 0.015 mM, 0.3 mM, and 0.6 mM increased the ratio of late phase to early phase BCSC apoptosis to 4, 5, or 15 times greater than that of control cells treated only with DMSO (Fig 6B).

## Discussion

In our *in silico* study, curcumin, rocaglamide, α-mangostin, 6-gingerol, 8-gingerol, 10-gingerol, and andrographolide were selected as natural dietary compounds for cancer treatment in humans [12–20]. According to Lipinski's Rule of Five, a candidate drug must not have a molecular weight of more than 500 Da, must have a number of hydrogen bond donors that is less than or equal to 5, must have a number of hydrogen bonds—both donor and acceptor—that is less than 10, must have a log P value of less than 5, and must have a total polar surface area (TPSA) of less than 140Å [24]. These rules are generally used not only to predict the absorption of compounds but also to evaluate the overall druglikeness property. Following a molecular docking simulation, andrographolide and rocaglamide were considered as hit compounds for further molecular interaction analysis with survivin.

Based on the ∆G binding and pKi value, rocaglamide has the higher affinity with survivin when compared to andrographolide. However, andrographolide has lesser number of hydrogen bonds but is specific to the phosphorylation site, Thr34 residue, as well as having more electrostatic interactions than rocaglamide. Previous studies have suggested that electrostatic interactions are preferred over hydrogen bonds in drug-receptor interactions [23–25]. Therefore, we suggest that andrographolide is a better candidate to be used as the lead compound to inhibit survivin activity.

The results from the *in silico* study were verified by an *in vitro* study using human BCSCs. Our *in vitro* results revealed that andrographolide at concentrations of 0.3 and 0.6 mM has a significant cytotoxic effect on human BCSCs. Although the effect of andrographolide on the BCSCs is quite remarkable, its $CC_{50}$ value of 0.32 mM in the CD24-/CD44+ BCSCs is higher than the doses of most marketed drugs against breast cancer that are effective at a lower concentration. Nevertheless, the most anti-cancer that are effective at a lower concentration are targeting the major population of breast cancer cells without considering the presence of BCSCs. In this study, we specifically targeted the side population of breast cancer cells–BCSCs–that are known to be less sensitive to anti-cancer than the non-BCSCs [36, 37]. We indicated that the $CC_{50}$ value of andrographolide in BCSCs is significantly higher than that in non-BCSCs such as human CD24-/CD44- breast cancer cells and human MCF-7 breast cancer cells (S2 Fig). Hence, the BCSCs need higher doses of anti-cancer than the major part of breast cancer cells which are not stem cells. Despite the high dose of andrographolide needed to target survivin in BCSCs, the present data showed that this dose was still non-toxic to the normal stem cells (Fig 4). We found that andrographolide is not toxic to MSCs derived from dental pulp even at $CC_{90}$ (0.6 mM) of andrographolide in human BCSCs.

The drug-resistance of BCSCs may be due to the higher expression of survivin in the human BCSCs compared to their counterpart non-BCSCs, as indicated in our previous data [7]. Our supporting data also revealed that the proliferation rate of BCSCs was inhibited when treated with 100 nM YM155, a survivin inhibitor [S3 Fig]. This suggests that the inhibition of survivin has a determinant role in the anti-cancer activity of andrographolide. Through molecular docking simulation, we further demonstrate that andrographolide could bind not only to the Thr34 active site, but also to other amino acid residues (5, 102, 105, 106, 109, 110, 113; Table 5). Unlike Thr34, which is well-known to be phosphorylated for the activation of survivin, the role of the seven other amino acid residues in survivin activation still needs to be investigated. We also confirmed that andrographolide significantly reduced the Thr34-phosphorylated survivin in BCSCs *in vitro*, indicating that andrographolide inhibits survivin activation. Furthermore, previous studies have shown that phosphorylation of survivin at Thr 117 by Aurora-B kinase plays an important role during mitosis [38, 39]. The action of andrographolide in this region may disrupt the interaction between survivin and microtubules, resulting in the loss of the anti-apoptotic function of survivin and increased caspase-3 activity during mitosis. This may explain why andrographolide inhibits not only the Thr34 phosphorylated survivin, but also reduces the survivin level as indicated by the results of ELISA and immunoblot assay in this study. Alternatively, the reduced total survivin level found in our *in vitro* study may be due to the failure of survivin activation, which induced ubiquitination leading to protein degradation [40].

Numerous studies have shown that survivin blocks the formation of apoptosomes, leading to the inhibition of procaspase-9 activation, and that it might directly inhibit caspase-9 or caspase-3 [41–43]. In addition, there is a possibility that andrographolide not only interacts with survivin, but also with active (cleaved) caspase-9 and procaspase-3 with a similar affinity to that which it has with the Thr34 residue or the other binding sites of survivin. Our molecular docking simulation indicated that the presence of andrographolide would affect the affinity of

protein-protein interactions between survivin, caspase-9, and caspase-3. The high affinity between survivin and caspase-9, as well as between survivin and caspase-3, could be reduced by the presence of andrographolide, thereby inhibiting the anti-apoptotic property of survivin. Conversely, andrographolide markedly increases the affinity of active caspase-9 and procaspase-3. This effect might be correlated with the binding of andrographolide to active initiator caspase-9, which then induces the activation of procaspase-3 to execute the intrinsic apoptosis pathway. It should be noted that the binding of andrographolide with either caspase-9 or caspase-3 requires higher energy than the binding of survivin to each protein, therefore, this binding cannot occur if survivin has bound to both proteins. Although andrographolide has been previously shown to play a functional role in inhibiting intrinsic apoptosis through blocking caspase activation [17], the role of andrographolide in survivin inactivation remained unclear until now.

Another anti-apoptotic mechanism of survivin that should be considered is through its direct interaction with Smac/DIABLO in the cytosol during interphase [44]. Smac/DIABLO is a mitochondrial protein that potentiates apoptosis by activating caspase in the cytochrome c/Apaf-1/caspase-9 pathway [45]. In our *in silico* study, we confirm that survivin directly binds to Smac/DIABLO (S8 Fig) which in turn may prevent caspase activation and inhibit apoptosis [44]. Moreover, the presence of andrographolide decreased the binding affinity of survivin and Smac/DIABLO *in silico* (S5 Table) and reduced survivin level *in vitro*, allowing Smac/DIABLO to promote caspase-activated apoptosis, as shown by the Annexin V-FITC result (Fig 6). However, the effect of andrographolide treatment on caspase activation by Smac/DIABLO should be further elucidated.

As well as analyzing the effect of andrographolide on survivin protein levels and activation, we also examined the expression of survivin, caspase-9, and caspase-3 mRNA in order to investigate cells' adaptive responses to the effects of andrographolide treatment on the expression regulation of respective genes. In the present study, survivin mRNA expression in BCSCs treated with 0.6 mM andrographolide was increased, possibly in response to the reduction of total and phosphorylated survivin levels. Caspase-9 mRNA expression was not affected by andrographolide treatment, signifying that the caspase-9 protein level in BCSCs was sufficient for cleavage of caspase-3 and apoptosis activation, since survivin could not bind to caspase-9 anymore. This assumption was supported by immunoblot assay that revealed the increase caspase-9 protein levels after andrographolide treatment. As a response of survivin inhibition by andrographolide, caspase-3 gene expression—an executor protein of apoptosis—was significantly up-regulated at andrographolide concentrations of less than $CC_{50}$ (0.32 mM). At andrographolide concentrations greater than $CC_{50}$ this effect disappeared, the caspase-3 protein expression was still slightly increased to induce both the early and late apoptosis phases in a concentration-dependent manner until $CC_{90}$ (0.6 mM) as shown by Annexin-V/PI apoptosis assay. This phenomenon might be attributable to other programmed cell death pathways which could also be induced by andrographolide, such as the extrinsic apoptosis pathway or autophagy [46, 47]. Research conducted by Tan *et al.* also reported that 14-deoxy-11,12-dehydro andrographolide (14-DDA) can induce autophagy in breast cancer cells T47D [47]. Cell death caused by autophagy is similar to apoptosis in terms of the disruption of cell membrane integrity and phosphatidylserine externalization, which can also be detected by Annexin-V/PI apoptosis assay. A previous study has also reported that andrographolide compounds increased the activation of caspase-3, 8, and 9 in hepatocellular carcinoma [48]. In addition, another study by Das *et al.* showed that andrographolide induced apoptosis and autophagy in leukemia [49].

Altogether, we emphasize the effectiveness of andrographolide in increasing the caspase activation cascade of the intrinsic apoptosis pathway (Fig 7) and propose that the binding of andrographolide to survivin is a critical aspect of this mechanism.

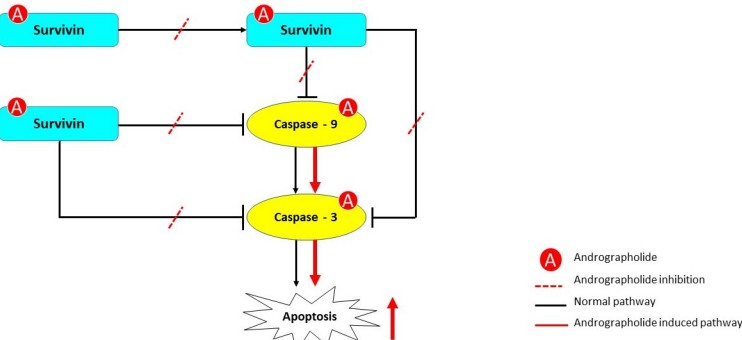

**Fig 7. Proposed mechanism of andrographolide on the intrinsic apoptosis pathway.** Andrographolide interacts with survivin, phosphorylated survivin, caspase-9, and caspase-3. Andrographolide treatment could inhibit the phosphorylation of survivin and the binding of survivin and p-survivin to caspase-9 and caspase-3, as shown by the dashed red line. As a consequence, intrinsic apoptosis could be induced through activation of caspase-9 and caspase-3, as shown by the continuous red line in the pathway.

## Conclusion

We examined the anti-cancer potential of andrographolide, a natural active compound, through its *in silico* molecular interactions with survivin, caspase-9, and caspase-3 and the increased intrinsic apoptosis activity of human BCSCs *in vitro*. We suggest that the binding of andrographolide to survivin, either at the Thr34 phosphorylation site or other active sites, is a critical part of the effect of andrographolide. We also verified that andrographolide is not toxic to normal MSCs, indicating that andrographolide could be considered as a novel adjuvant anti-cancer treatment targeting survivin, particularly in human BCSCs.

## Supporting information

**S1 Fig. Physicochemical data of seventh active compounds.** Physicochemical data obtained using SWISSADME server.
(DOCX)

**S2 Fig. Cytotoxic activity of andrographolide in human CD24-/CD44- breast cancer cells (non-BCSCs), and human MCF-7 breast cancer cell line compared to that in human BCSCs.** A total of 100.000 cells were treated with 0.075, 0.15, 0.3, or 0.6 mM andrographolide for 24 hours. Data are presented as mean ± standard deviation (SD) and analyzed using Student's t-test. The significance levels were shown as **p<0.01 compared to CD24-/CD44+ cells.
(DOCX)

**S3 Fig. Proliferation rate of human CD24-/CD44+ BCSCs with 100nM YM155, a survivin inhibitor.** A total of 100.000 cells were treated with 100nM YM155, a survivin inhibitor, at Day 0, 2, and 4. Cells were then harvested and counted every day. Data are presented as mean ± standard deviation (SD) and analyzed using Student's t-test. The statistical significance levels were shown as **p<0.01 and *p<0.05 compared to its respective control without YM155.
(DOCX)

**S4 Fig. Melting curve qRT-PCR 18S rRNA, survivin, caspase-9, and caspase-3.**
(DOCX)

**S5 Fig. Statistical analysis of qRT-PCR and ELISA data using SPSS software version 26.**
(DOCX)

**S6 Fig. Immunoblot of survivin, caspase-9, caspase-3, and beta actin.** Membrane was first used for immunoblotting with anti caspase-3 antibody. Then, the membrane was horizontally cut into 2 parts at 30 kDa level and stripped for immunoblotting with anti-beta actin, anti-survivin, anti-caspase-9, and anti-phospho survivin.
(DOCX)

**S7 Fig. Data of apoptosis analysis of andrographolide-treated BCSCs using flow cytometry.**
(Output Data from flow cytometry Instrument).
(DOCX)

**S8 Fig. Molecular docking between survivin, Smac/DIABLO, and andrographolide using autodock software version 4.2.**
(DOCX)

**S9 Fig.**
(DOCX)

**S1 Table. Energy values obtained in molecular docking calculations with survivin.**
(DOCX)

**S2 Table.**
(DOCX)

**S3 Table. qRT-PCR data of survivin, Caspase-9, Caspase-3 mRNA expression levels in BCSCs treated with various concentrations of andrographolide.**
(DOCX)

**S4 Table. ELISA data of total and phosphorylated survivin protein levels in BCSCs treated with various andrographolide.**
(DOCX)

**S5 Table. Binding energy of survivin and Smac/DIABLO in the presence of andrographolide.**
(DOCX)

## Acknowledgments

The authors would like to thank Dr. Fadilah Fadilah, S.Si, M.Si and Dr. Firdayani, M.Farm for their insight and assistance regarding molecular docking analysis, Karina Teja Putri for her excellent suggestions regarding English editing, as well as Prof. Endang W. Bachtiar for providing us the MSCs from dental pulp.

## Author Contributions

**Conceptualization:** Septelia Inawati Wanandi, Melva Louisa, Agung Eru Wibowo.

**Data curation:** Septelia Inawati Wanandi, Agus Limanto, Elvira Yunita, Resda Akhra Syahrani, Sekar Arumsari.

**Formal analysis:** Septelia Inawati Wanandi, Agus Limanto, Elvira Yunita, Melva Louisa, Agung Eru Wibowo.

**Funding acquisition:** Septelia Inawati Wanandi.

**Investigation:** Septelia Inawati Wanandi, Agus Limanto, Elvira Yunita, Sekar Arumsari.

**Methodology:** Septelia Inawati Wanandi, Agus Limanto, Elvira Yunita, Resda Akhra Syahrani, Melva Louisa, Sekar Arumsari.

**Project administration:** Resda Akhra Syahrani.

**Resources:** Agung Eru Wibowo.

**Software:** Agus Limanto.

**Supervision:** Septelia Inawati Wanandi, Melva Louisa, Agung Eru Wibowo.

**Validation:** Septelia Inawati Wanandi, Agus Limanto, Resda Akhra Syahrani, Melva Louisa, Agung Eru Wibowo, Sekar Arumsari.

**Visualization:** Agus Limanto, Elvira Yunita, Resda Akhra Syahrani.

**Writing – original draft:** Septelia Inawati Wanandi, Agus Limanto, Elvira Yunita.

**Writing – review & editing:** Septelia Inawati Wanandi, Melva Louisa, Agung Eru Wibowo, Sekar Arumsari.

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
