## [Decision Letter · Decision Letter 0]

1 Apr 2020

PONE-D-20-03995

In silico and in vitro studies on the anti-cancer activity of andrographolide targeting survivin in human breast cancer stem cells

PLOS ONE

Dear Prof. Dr.rer.physiol. dr. Wanandi,

Thank you for submitting your manuscript to PLOS ONE. After careful consideration, we feel that it has merit but does not fully meet PLOS ONE’s publication criteria as it currently stands. Therefore, we invite you to submit a revised version of the manuscript that addresses the points raised during the review process.

As noted in the reviewers comments below, additional experimental studies would be needed to robustly support the conclusions. In particular, authors are requested to include data regarding activation of caspases 3 and 9 by either western blotting or by ApoAlert caspase profiling assay. 

We would appreciate receiving your revised manuscript by May 16 2020 11:59PM. To enhance the reproducibility of your results, we recommend that if applicable you deposit your laboratory protocols in protocols.io, where a protocol can be assigned its own identifier (DOI) such that it can be cited independently in the future. For instructions see: http://journals.plos.org/plosone/s/submission-guidelines#loc-laboratory-protocols

We look forward to receiving your revised manuscript.

Kind regards,

Arun Rishi, Ph.D.

Academic Editor

PLOS ONE

Journal Requirements:

1. Please provide additional details regarding consent sought from patients to obtain the primary breast cancer stem cells and mesenchymal stem cells used in this study.

2.  In the ethics statement in the Methods and online submission information, please ensure that you have specified (a) whether consent was informed and (b) what type you obtained (for instance, written or verbal, and if verbal, how it was documented and witnessed).

3.  On the contrary, if these cells are commercially available cell lines, please include this information and state the specific name of the cell lines.

4. Please provide additional information about each of the cells used in this work, including any quality control testing procedures (mycoplasma testing). For more information, please see http://journals.plos.org/plosone/s/submission-guidelines#loc-cell-lines.

5. To comply with PLOS ONE submission guidelines, in your Methods section, please provide additional information regarding your statistical analyses. For more information on PLOS ONE's expectations for statistical reporting, please see https://journals.plos.org/plosone/s/submission-guidelines.#loc-statistical-reporting.

6. Thank you for including the following funding information in your acknowlegdements section; "This work was funded by the PDUPT Grant (NKB-1564/UN2.R3.1/HKP.05.00/2019) from the Ministry of Research, Technology and Higher Education of the Republic of Indonesia. The funders had no role in study design, data collection and analysis, decision to publish, or preparation of the manuscript. "

7. 

We note that you have indicated that data from this study are available upon request. PLOS only allows data to be available upon request if there are legal or ethical restrictions on sharing data publicly. For more information on unacceptable data access restrictions, please see http://journals.plos.org/plosone/s/data-availability#loc-unacceptable-data-access-restrictions.

8. In the Methods section, please provide the primer sequences and source of the primers for the 18srRNA mRNA amplifications.

9. In the Methods section, please provide the product number and lot number of the Andrographolide purchased from Sigma-Aldrich for this study.

Reviewers' comments:

Reviewer's Responses to Questions

**Comments to the Author**

1. Is the manuscript technically sound, and do the data support the conclusions?

Reviewer #1: Yes

Reviewer #2: Partly

2. Has the statistical analysis been performed appropriately and rigorously? 

Reviewer #1: I Don't Know

Reviewer #2: Yes

3. Have the authors made all data underlying the findings in their manuscript fully available?

Reviewer #1: Yes

Reviewer #2: Yes

4. Is the manuscript presented in an intelligible fashion and written in standard English?

Reviewer #1: Yes

Reviewer #2: Yes

5. Review Comments to the Author

Reviewer #1: Septelia Inawati Wanandi et al. reports that andrographolide anti-cancer activity by targeting survivin in human breast cancer stem cells. The study is supported by the previous study that andrographolide suppresses the proliferation of pancreatic cancer cells via downregulating surviving (G.Q. Bao, B.Y. Shen, C.P. Pan, Y.J. Zhang, M.M. Shi, C.H. Peng, Andrographolide causes apoptosis via inactivation of STAT3 and Akt and potentiates antitumor activity of gemcitabine in pancreatic cancer, Toxicol Lett 222 (2013) 23-35.). However, it is not known the effects of andrographolide in breast cancer stem cells. Overall, this study is well designed and the results obtained support the conclusions drawn. I support this manuscript to be accepted with the following revisions.

Major comments:

1. Fig. 4C is repeated.

2. The authors are recommended to investigate the anti-cancer capacity of andrographolide and survivin inhibition by andrographolide in other non-BCSCs. It is better to compare the anti-cancer activity of andrographolide in BCSCs and non-BCSCs.

3. For Fig.5B, please clearly label the drug concentration used in the study.

4. In the legend of Fig.5B (“BCSC morphology treated with 0.06 mM andrographolide”). Please make sure the drug concentration is corrected.

5. Fig.4 and Fig.5 should be merged into one Figure.

6. For Fig.6, the authors are strongly recommended to detect survivin and phosphorylated survivin protein level by using immunoblotting assay.

7. For Fig.7, the authors need to further measure the activation of apoptosis-associated proteins (e.g. PARP, caspase-3, caspase-9, and so on) by immunoblotting.

8. To make sure the determinant role of survivin inhibition by andrographolide in its anti-cancer activity in breast cancer stem cells, the authors need to show the proliferation rates of BCSCs and surviving-KO/KD BCSCs in their system.

9. The authors need to clarify the statistical analysis in the Materials and Methods section.

Reviewer #2: 1. The authors have studied the expression levels of caspase 3, caspase 9 and survivin at the mRNA level. After translation in the cytoplasm ,the proteins undergo a lot of post translational modifications. So, my suggestion would be to please check the expression levels of caspase 9, caspase 3 and survivin at the protein levels using standard western blot or flow cytometry protocols.

2. The authors have selected 7 natural compounds from medicinal plants, curcumin, rocaglamide, α-mangostin, 6-gingerol, 8-gingerol, 10-gingerol, and Andrographolide, all of which have anticancer properties. However the authors did not mention why they selected these specific compounds, given the fact that there are a lot of other available natural compounds, having anticancer potential.

3. Apoptosis is a programmed cell death process involving the participation of lot of proteins. Survivin functions by binding to pro-apoptotic proteins SMAC/DIABLO, which in turn prevents caspase activation. So, studying the expression levels of these proteins at the genetic and protein levels are also required.

4. In the graphical abstract part replace the second caspase 9 with caspase 3.

5. The effect of Andrographolide on breast cancer stem cells is quite remarkable but IC50 value of 320μM is pretty high dose on cancer cells, considering the fact that marketed drugs against breast cancer are effective at 10-20μM concentration. What could be the other possible reasons of selecting such a compound for anticancer activity, other than the fact that its non toxic to normal stem cells?

6. In the legend fig 5B ‘BCSC morphology treated with 0.06 mM andrographolide’, recheck if it is 0.06mM or 0.6mM?

7. Include the statistical significances in fig.7B

6. PLOS authors have the option to publish the peer review history of their article (what does this mean?). If published, this will include your full peer review and any attached files.

Reviewer #1: Yes: Wei Wei

Reviewer #2: No

---

## [Author Response · Author response to Decision Letter 0]

6 Aug 2020

PONE-D-20-03995

In silico and in vitro studies on the anti-cancer activity of andrographolide targeting survivin in human breast cancer stem cells

Jakarta, July 30th, 2020

Dear Arun Rishi, Ph.D.

Academic Editor

PLOS ONE

First of all, I would like to extend our sincere thanks to the PLOS ONE academic editor and reviewers for the fruitful comments and also for giving us the opportunity to revise our manuscript in this most difficult time due to pandemic situation. Herewith, I kindly send you the responds to each point raised by the academic editor and reviewers in the following Tables. 

Responses to Academic Editor’s comments

1. Please provide additional details regarding consent sought from patients to obtain the primary breast cancer stem cells and mesenchymal stem cells used in this study. 

Response: We have provided the following details: “Breast cancer tissue has been collected from a triple negative breast cancer patient underwent surgery. Written informed consent form has been signed by the patient prior to collecting the specimen. Mesenchymal stem cells (MSCs) were obtained from Prof. Endang Winiati Bachtiar (Laboratory of Oral Biology Faculty of Dentistry, Universitas Indonesia) who has isolated the MSCs from dental pulp of a patient underwent tooth extraction with a written informed consent signed by the patient and approved by the Ethics Commission of Research and Community Service Institution at Bogor Agricultural Institute (No. 05-2015 IPB). This study was conducted according to the ethical principles.” 

Page 5-6; Line 136-144.

2. In the ethics statement in the Methods and online submission information, please ensure that you have specified (a) whether consent was informed and (b) what type you obtained (for instance, written or verbal, and if verbal, how it was documented and witnessed). 

3. On the contrary, if these cells are commercially available cell lines, please include this information and state the specific name of the cell lines.

Response: In this study, we did not use any commercially available cell lines -

4. Please provide additional information about each of the cells used in this work, including any quality control testing procedures (mycoplasma testing). For more information, please see http://journals.plos.org/plosone/s/submission-guidelines#loc-cell-lines.

Response: Both BCSCs and MSCs used in this study are established cells that have been utilized in the previous studies (References has been added). The mycoplasma testing resulted in a negative compared to the positive control.

The following information has been added in the manuscript: “Human BCSCs (CD24-/CD44+) have been isolated from the primary culture of breast cancer cells (Registered patent from the General Directorate of Intellectual Property Right, Ministry of Law and Human Right, Republic of Indonesia No. IDP00056854) and characterized for their pluripotency and tumorigenicity (Registered patent from the General Directorate of Intellectual Property Right, Ministry of Law and Human Right, Republic of Indonesia No. IDP000060309). The human BCSCs and MSCs have been used in the previous studies [33-34].”

“The MSCs has been utilized in the previous study [33].”

Page 6 Line 147-152 and Page 6 Line 143.

5. To comply with PLOS ONE submission guidelines, in your Methods section, please provide additional information regarding your statistical analyses. For more information on PLOS ONE's expectations for statistical reporting, please see https://journals.plos.org/plosone/s/submission-guidelines.#loc-statistical-reporting.

Response: We have added the following information:

Statistical analysis

“Data were analyzed using SPSS software version 26. Data is presented as mean � standard deviation (SD) of three independent experiments with three replicates. Coefficient of variation (COV) is calculated as a percentage ratio of the SD to the mean of expression level. The COV value expresses the homogeneity of data distribution. Normality tests performed on all experiment data were Kolmogorof-Smirnov test and the Levene homogeneity test. Since the data distribution is normal and homogenous, the analysis was continued with the parametric test of one-way analysis of unpaired variance (one-way ANOVA, unrelated) followed by the Tuckey test to show the differences between groups”. Page 9; Line 226-234.

6. Thank you for including the following funding information in your acknowlegdements section; "This work was funded by the PDUPT Grant (NKB-1564/UN2.R3.1/HKP.05.00/2019) from the Ministry of Research, Technology and Higher Education of the Republic of Indonesia. The funders had no role in study design, data collection and analysis, decision to publish, or preparation of the manuscript. "

Response: Thank you for correcting this mistake. We removed any funding-related text from the manuscript.

Please update our Funding Statement as follows:

"This work was funded by the PDUPT Grant (NKB-1564/UN2.R3.1/HKP.05.00/2019) from the Ministry of Research, Technology and Higher Education of the Republic of Indonesia. The funders had no role in study design, data collection and analysis, decision to publish, or preparation of the manuscript."

If there are no restrictions, please upload the minimal anonymized data set necessary to replicate your study findings as either Supporting Information files or to a stable, public repository and provide us with the relevant URLs, DOIs, or accession numbers. 

For a list of acceptable repositories, please see http://journals.plos.org/plosone/s/data-availability#loc-recommended-repositories.

Response: The minimal anonymized data set necessary to replicate our study findings can be found in Supporting Information files.

Please update our data availability as follows: “All relevant data are within the manuscript and its Supporting Information files.”

Page 21; Line 519

8. In the Methods section, please provide the primer sequences and source of the primers for the 18srRNA mRNA amplifications. 

Response: We have provided the sources of all primers used in this study, as follows:

“The primers for survivin, caspase-9, caspase-3, and 18S rRNA cDNA amplification were designed using the Primer-Blast program and NCBI Gene Bank [NM_001168.2, NM_01278054.1, NM_004346.4, and NC_000021.9, respectively].”

Page 7; Line 182-184.

9. In the Methods section, please provide the product number and lot number of the Andrographolide purchased from Sigma-Aldrich for this study. 

Response: We have provided the following information for andrographolide:

“Product No. 365645; Lot No. MKBN8939V” Page 6; Line 161.

Responses to Reviewer I’s comments

1. Fig. 4C is repeated 

Response: We have deleted the Fig. 4C.

Page 14-15; Line 326-333.

2. The authors are recommended to investigate the anti-cancer capacity of andrographolide and survivin inhibition by andrographolide in other non-BCSCs. It is better to compare the anti-cancer activity of andrographolide in BCSCs and non-BCSCs. 

Response: We have compared the anti-cancer activity of andrographolide in BCSCs and non-BCSCS. The following statement has been added in the manuscript.

“We indicated that the CC50 value of andrographolide in BCSCs is significantly higher than that in non-BCSCs such as human CD24-/CD44- breast cancer cells and human MCF-7 breast cancer cells (S2 Figure).” 

Page 18; Line 420-422.

3. For Fig.5B, please clearly label the drug concentration used in the study. 

Response: Thank you for your correction. We have revised the drug concentration. 

In Figure 4D we have added:

“BCSC morphology treated with 0.6 mM andrographolide”.

Page 15; Line 333.

4. In the legend of Fig.5B (“BCSC morphology treated with 0.06 mM andrographolide”). Please make sure the drug concentration is corrected. 

Response: Thank you for your correction. We have revised the drug concentration. It should be 0.6 mM. 

5. Fig.4 and Fig.5 should be merged into one Figure. 

Response: We have merged Fig. 4 and Fig. 5 into one Figure.

Page 14-15; Line 326-333.

6. For Fig.6, the authors are strongly recommended to detect survivin and phosphorylated survivin protein level by using immunoblotting assay. We have detected the protein level of survivin and phosphorylated survivin using immunoblotting assay.

Response: The data regarding this immunoblotting assay has been added in the manuscript. 

Page 8-9; Line 204-214

Page 15-16; Line 356-360

Page 16; Line 367-368

Page 19; Line 443-445

7. For Fig.7, the authors need to further measure the activation of apoptosis-associated proteins (e.g. PARP, caspase-3, caspase-9, and so on) by immunoblotting. 

Response: We have performed the immunoblotting assay to demonstrate the activation of caspase-9 and caspase-3.

The data regarding this immunoblotting assay has been added in the manuscript.

Page 8-9; Line 204-214

Page 16; Line 360-361

Page 16; Line 367-368

Page 20; Line 483-484 and Line 487-490

8. To make sure the determinant role of survivin inhibition by andrographolide in its anti-cancer activity in breast cancer stem cells, the authors need to show the proliferation rates of BCSCs and surviving-KO/KD BCSCs in their system.

Response: We have performed the experiments to show the proliferation rates of BCSCs and survivin-inhibited BCSCs (using YM155 as a survivin inhibitor), as indicated by their doubling times.

The following result has been added in the manuscript:

“Our supporting data also revealed that the proliferation rate of BCSCs was inhibited when treated with 100 nM YM155, a survivin inhibitor [S3 Figure]. This suggests that the inhibition of survivin has a determinant role in the anti-cancer activity of andrographolide.” 

Page 18; Line 430-432

9. The authors need to clarify the statistical analysis in the Materials and Methods section.

Response: We have added the following information:

Statistical analysis

“Data were analyzed using SPSS software version 26. Data is presented as mean � standard deviation (SD) of three independent experiments with three replicates. Coefficient of variation (COV) is calculated as a percentage ratio of the SD to the mean of expression level. The COV value expresses the homogeneity of data distribution. Normality tests performed on all experiment data were Kolmogorof-Smirnov test and the Levene homogeneity test. Since the data distribution is normal and homogenous, the analysis was continued with the parametric test of one-way analysis of unpaired variance (one-way ANOVA, unrelated) followed by the Tuckey test to show the differences between groups.”

Page 9; Line 226-234

Responses to Reviewer II’s comments

1 The authors have studied the expression levels of caspase 3, caspase 9 and survivin at the mRNA level. After translation in the cytoplasm ,the proteins undergo a lot of post translational modifications. So, my suggestion would be to please check the expression levels of caspase 9, caspase 3 and survivin at the protein levels using standard western blot or flow cytometry protocols. 

Response: We have performed the immunoblotting assay to demonstrate the activation of caspase-9 and caspase-3.

The data regarding this immunoblotting assay has been added in the manuscript.

Page 8-9; Line 204-214

Page 15-16; Line 356-361

Page 16; Line 367-368

Page 19; Line 443-445

Page 20; Line 483-484 and Line 487-490

2 The authors have selected 7 natural compounds from medicinal plants, curcumin, rocaglamide, α-mangostin, 6-gingerol, 8-gingerol, 10-gingerol, and Andrographolide, all of which have anticancer properties. However the authors did not mention why they selected these specific compounds, given the fact that there are a lot of other available natural compounds, having anticancer potential. 

Response have added the following informations:

“These selected active compounds are found among medicinal plants such as Curcuma domestica, Curcuma xantorrhiza, Aglaia sp., Andrographis paniculata and Garcinia mangostana that are endemic in Southeast Asian countries, including Indonesia [9-11]. Therefore, utilization of these compounds for anticancer offers a potential benefit, both economically and clinically.”

Page 3; Line 76-80

3 Apoptosis is a programmed cell death process involving the participation of lot of proteins. Survivin functions by binding to pro-apoptotic proteins SMAC/DIABLO, which in turn prevents caspase activation. So, studying the expression levels of these proteins at the genetic and protein levels are also required. 

Response: The following information have been added in the manuscript:

“Another anti-apoptotic mechanism of survivin that should be considered is through its direct interaction with Smac/DIABLO in the cytosol during interphase [44]. Smac/DIABLO is a mitochondrial protein that potentiates apoptosis by activating caspase in the cytochrome c/Apaf-1/caspase-9 pathway [45]. In our in silico study, we confirm that survivin directly binds to Smac/DIABLO (S8 Figure) which in turn may prevent caspase activation and inhibit apoptosis [44]. Moreover, the presence of andrographolide decreased the binding affinity of survivin and Smac/DIABLO in silico (S5 Table) and reduced survivin level in vitro, allowing Smac/DIABLO to promote caspase-activated apoptosis, as shown by the Annexin V-FITC result (Figure 6). However, the effect of andrographolide treatment on caspase activation by Smac/DIABLO should be further elucidated.”

Page 19; Line 465-474

4 In the graphical abstract part replace the second caspase 9 with caspase 3.

Response: Thank you for your correction. We have replaced the second caspase-9 with caspase-3 in Figure 7. Figure 7

5 The effect of Andrographolide on breast cancer stem cells is quite remarkable but IC50 value of 320μM is pretty high dose on cancer cells, considering the fact that marketed drugs against breast cancer are effective at 10-20μM concentration. What could be the other possible reasons of selecting such a compound for anticancer activity, other than the fact that its non toxic to normal stem cells? 

Response: We agree with the Reviewer’s comment regarding the high CC50 value of BCSCs.

We have added the following information to respond to this comment.

“Although the effect of andrographolide on the BCSCs is quite remarkable, its CC50 value of 0.32 mM in the CD24-/CD44+ BCSCs is higher than the doses of most marketed drugs against breast cancer that are effective at a lower concentration. Nevertheless, the most anti-cancer that are effective at a lower concentration are targeting the major population of breast cancer cells without considering the presence of BCSCs. In this study, we specifically targeted the side population of breast cancer cells – BCSCs – that are known to be less sensitive to anti-cancer than the non-BCSCs [36-37]. We indicated that the CC50 value of andrographolide in BCSCs is significantly higher than that in non-BCSCs such as human CD24-/CD44- breast cancer cells and human MCF-7 breast cancer cells (S2 Figure). Hence, the BCSCs need higher doses of anti-cancer than the major part of breast cancer cells which are not stem cells. Despite the high dose of andrographolide needed to target survivin in BCSCs, the present data showed that this dose was still non-toxic to the normal stem cells (Figure 4). We found that andrographolide is not toxic to MSCs derived from dental pulp even at CC90 (0.6 mM) of andrographolide in human BCSCs.” 

Page 18; Line 414-427

6 In the legend fig 5B ‘BCSC morphology treated with 0.06 mM andrographolide’, recheck if it is 0.06mM or 0.6mM?

Response: Thank you for your correction. 

We have revised the drug concentration. 

In Figure 4D we have added:

“BCSC morphology treated with 0.6 mM andrographolide”.

Page 15; Line 333

7 Include the statistical significances in fig.7B

Response: Thank you for your suggestion. 

For apoptosis assay using flow cytometry, we pooled triplicates of each sample to obtain enough cells after andrographolide treatment. Andrographolide remarkably reduced viable cell number (up to ca. 90% with 0.6 mM andrographolide; Fig. 4A). Therefore, we could not determine the statistical significances in the apoptosis result. 

Thank you for considering this manuscript to be published in your highly reputable journal.

Sincerely yours,

Prof. Dr.rer.physiol. dr. Septelia Inawati Wanandi

Department of Biochemistry and Molecular Biology, 

Faculty of Medicine, Universitas Indonesia, Indonesia

Email: septelia.inawati@ui.ac.id; septelia.inawati@gmail.com

Phone: +6221-3910734

---

## [Decision Letter · Decision Letter 1]

18 Sep 2020

In silico and in vitro studies on the anti-cancer activity of andrographolide targeting survivin in human breast cancer stem cells

PONE-D-20-03995R1

Dear Dr. Wanandi,

We’re pleased to inform you that your manuscript has been judged scientifically suitable for publication and will be formally accepted for publication once it meets all outstanding technical requirements.

Kind regards,

Arun Rishi, Ph.D.

Academic Editor

PLOS ONE

Additional Editor Comments (optional):

Reviewers' comments:

Reviewer's Responses to Questions

**Comments to the Author**

1. If the authors have adequately addressed your comments raised in a previous round of review and you feel that this manuscript is now acceptable for publication, you may indicate that here to bypass the “Comments to the Author” section, enter your conflict of interest statement in the “Confidential to Editor” section, and submit your "Accept" recommendation.

Reviewer #1: All comments have been addressed

2. Is the manuscript technically sound, and do the data support the conclusions?

Reviewer #1: Yes

3. Has the statistical analysis been performed appropriately and rigorously? 

Reviewer #1: Yes

4. Have the authors made all data underlying the findings in their manuscript fully available?

Reviewer #1: Yes

5. Is the manuscript presented in an intelligible fashion and written in standard English?

Reviewer #1: Yes

6. Review Comments to the Author

Reviewer #1: (No Response)

7. PLOS authors have the option to publish the peer review history of their article (what does this mean?). If published, this will include your full peer review and any attached files.

Reviewer #1: **Yes: **Wei Wei

---

## [Editor Report · Acceptance letter]

30 Sep 2020

PONE-D-20-03995R1 

*In* *silico* and *in*
*vitro* studies on the anti-cancer activity of andrographolide targeting survivin in human breast cancer stem cells 

Dear Dr. Wanandi:

I'm pleased to inform you that your manuscript has been deemed suitable for publication in PLOS ONE. Congratulations! Your manuscript is now with our production department. 

Kind regards, 

on behalf of

Prof Arun Rishi 

Academic Editor

PLOS ONE